# Exploring Public Perceptions and Disposal Procedures in the Development of a Comprehensive End-of-Life Vehicle Regulation in Malaysia: A Pilot Study

Hasani Mohd Ali [1] , Vladimir Simic [2] , Charli Sitinjak [3,4,*] , Jady Zaidi Hassim [1] , Muhamad Helmi Md Said [1,*] , Rasyikah Md Khalid [1] , Grace Emmanuel Kaka [1] and Rozmi Ismail [3]

1  Faculty of Law, Universiti Kebangsaan Malaysia, Bangi 43600, Selangor, Malaysia
2  Faculty of Transport and Traffic Engineering, University of Belgrade, Vojvode Stepe 305, 110110 Belgrade, Serbia
3  Centre for Research in Psychology and Human Well-Being (PSiTra), Faculty of Social Sciences and Humanities, Universiti Kebangsaan Malaysia, Bangi 43600, Selangor, Malaysia
4  Faculty of Psychology, University Esa Unggul, West Jakarta City 11510, Indonesia
*  Correspondence: p112562@siswa.ukm.edu.my (C.S.); mhelmisaid@ukm.edu.my (M.H.M.S.)

**Abstract:** The growing global demand for new cars has resulted in a rise in end-of-life vehicles (ELV), both with usable and non-usable parts. Malaysia faces a particularly pressing issue of abandoned vehicles (AVs), as the country currently lacks comprehensive legislation for managing ELV. On the other hand, countries such as the US, Japan, Belgium, Korea, and China have well-established ELV management policies. In light of this, a pilot study assessed the public's perception and attitudes towards ELV issues and regulations in Malaysia. The study gathered participants' views on surrender, deregistration, disposal costs, incentives, recycling, and ELV regulations. The questionnaire used in the study was based on the findings from a previous report. The data were analysed using SPSS version 27, based on 40 interviews with diverse participants. The results of the study revealed three primary themes: "AV and its ELV", "ELV disposal procedures", and "ELV regulation and conflict resolution", with 14 sub-themes. The study adopts a regulatory approach towards ELV and AVs and highlights the importance of an effective voluntary surrender system in Malaysia as a step towards comprehensive ELV regulation. The findings of this pilot study provide valuable insights into the Malaysian public's views on ELV and can inform the development of more effective and comprehensive ELV legislation.

**Keywords:** end-of-life vehicle; abandoned vehicle; authorized automotive treatment facility; guidelines/regulations; recycling policy; public participation; Malaysia

## 1. Introduction

Over previous decades, rapid global development resulted in excessive pressures on limited natural resources. These developments saw an increased demand for vehicles [1]. These consumption demands were fuelled by consumers' desires for the latest launched products [2], and manufacturers' plans to discard obsolete products [3] led to the large production quantity of waste products straining the environment [4,5]. As a result, the requirement for long-term sustainable development is jeopardized [6]. Similarly, the linear economy model "make-take-use-dispose" [7] poses societal and environmental concerns. Hence, the need to adopt the reuse and recycle process for vehicles that have reached their end of life. This approach will help avoid resource scarcity and environmental damage [8,9]. As such, there is a need for public awareness of the harmful effects of end-of-life vehicles (ELVs) and the valuable resources that can be harnessed from this waste flow. This initiative will require clarity on the management process for these ELVs to achieve a sustainable society [10]. The most recent data shows that China's automobile production and sales

surpassed 28 million vehicles in 2017, setting a new world record [11]. The industry is still showing signs of growth in 2018, reaching a car ownership rate of 150 million vehicles and a used vehicle ownership rate of 9 million vehicles by 2020 [12]. In Malaysia, there is no centralized legislation governing ELVs. Instead, the laws are dispersed among numerous pieces of legislation, such as ACT 133, ACT 333, the recent 2020 Guidelines, and the Criminal Procedure Code, to meet the procedural requirements. Hence, comprehensive legislation needs to determine ELVs and the management process.

Over the years, the craving for the latest vehicles has increased worldwide. This trend has led to an increase in ELVs. ELVs have both beneficial and harmful components. Approximately 20 to 30 thousand parts of ELVs are made of precious reusable materials such as platinum, aluminium, lead, zinc, copper, iron, glass, rubber, textile, wires, and plastics [13]. On the other hand, components such as oils, fuels, refrigerants, brominated flame-retardants, and acid batteries can harm human health and the environment [14]. A vehicle can become an ELV in a variety of ways. It can be the consequence of an accident, a flood, or simply because it has worn out due to age and is no longer cost-effective to maintain and repair. Three primary sources of ELVs are abandoned vehicles (AVs), insurers, and voluntary surrenders [15]. Thus, tackling ELVs is the sine qua non to a healthy life and environment. When determining whether to classify a car as a total loss after an incident involving fire, flood, or an accident when repair costs would exceed the car's market value, insurance companies also use ELV [16].

Cars are abandoned when they can no longer be fixed due to a lack of spare parts or the owner's inability to pay for such repairs [17]. An abandoned car at an enforcement agency's depot, such as the police station, following a seizure used as evidence in a criminal prosecution, can result in an ELV. The acquisition of cheap imported used cars and poor road conditions are some causes of AVs [18]. With time, AVs soon become an eyesore. Thus, AVs have become an environmental and public health (or health-related) issue worldwide [17]. AVs have generated global debate, proposals, measures, and conventions to curb the menace [15].

The presence of abandoned vehicles (AVs) in the environment poses significant ecological risks, including soil and groundwater contamination, air pollution, depletion of valuable land resources, and aesthetic degradation of the visual environment. The leakage of toxic substances from AVs, such as lead–acid batteries, fuels, oils, coolants, and chemicals, can result in soil and water pollution, posing a threat to the local ecosystem, wildlife, and human health. Additionally, the accumulation of AVs in landfill sites and the release of hazardous air pollutants and greenhouse gases can exacerbate environmental degradation. Thus, it is imperative to develop and implement effective management and disposal practices to mitigate the ecological impacts of AVs [13].

The United Nations Convention on Biodiversity from October 2021 set forth objectives for eliminating all discarded automobiles from public spaces to safeguard the environment and the lives of its inhabitants [17]. Various nations and islands have taken many approaches to address the issue of AVs. For instance, the Torres Strait Islands government removed and, in some cases, deeply buried AVs by spending nearly AU $16 million on a grant. To pay for a yearlong program of no-cost collection of up to two cars per inhabitant, a US $12 fee was added to vehicle registration on the Big Island Hawai'i. On Kauai, a proposed legislative fee was introduced for rental vehicle registration to help finance the collection and processing of AVs. Similarly, owners of AVs in the US Virgin Islands face fines of more than $1000 if they do not dispose of them.

Similarly, Belgium specified 12–15 years to determine ELVs. In addition, it used legislation to distribute subsidies for less polluting cars or to levy favourable taxes on cleaner engines for recyclability and new car material composition [19]. Korea passed the Recycling of Electrical and Electronic Products and Automobile Resources Act 2008 to attain the goals of reuse and recovery of ELVs [20]. This Act makes everyone responsible for recycling ELV, including manufacturers, importers, dismantlers, shredders, ASR recyclers, and refrigerant gas processors [21]. Japan used forward and reversed logistics [22,23]

through legislative involvement, to successfully reuse and recycle ELVs as a management strategy (see Appendix A Summary of previous Studies on ELV).

On the other hand, in Malaysia, there are no clear standards for the proper management of ELVs. Hence, increasing the probability of relatively high AVs. These issues prompted this study. First, a focus group discussion was carried out in the pilot study, forming the basis of the questionnaire developed for this study. The purpose of the research is to determine the general awareness among the public of ELVs and AVs. It is also to understand why AVs are still rising despite the existing legal framework. In addition, it is essential to know why people abandon their vehicles and cannot surrender them. The study took cognizance of the difficulties encountered in the surrendering process. The study tried to determine the public perception of the management and disposal of ELVs, what should constitute ELVs, and what processes, if adopted on ELVs, would encourage voluntary surrenders and reduce the problem of AVs in Malaysia. The study examined public perception while also creating awareness of the dangers and benefits of ELVs for a sustainable society. This study hopes to instigate comprehensive legislation on ELV management for disposal and reuse. By the end of the study, it is hoped that government and enforcement officers will see the loopholes in the present legislation and why enforcement is not feasible.

This research is divided into four sections. The first is the introduction and overview of ELVs, as discussed. The second section describes the method used in the study. The data presentation and analysis form the third section, and the last section is the discussion and conclusion part of the study.

## 2. Review of ELV Regulation and Management in Malaysia

According to data from the Department of Road Transport [24], there were 31.2 million registered vehicles as of 31 December 2019, and overall automotive industry sales (total industry volume) in Malaysia were 529,234 units in 2020 (Ministry of Transport Malaysia Road Safety Programs and Agencies, n.d.). This implies an expected rise in the number of ELVs in Malaysia to a vehicle-to-population ratio of 1:2.25. The rise will, undoubtedly, increase the workload for local authorities in managing AVs until they can no longer be regulated and handled efficiently. The legal framework for ELV management in Malaysia is restricted to AVs alone and governed under Act 133, Act 333, and the 2020 ministerial guidelines. This legislation and specific provisions on AVs and their management are analysed and represented in Table 1.

Although Table 1 indicates the existence of explicit provisions for the management of abandoned vehicles (AVs), recent data reveals a concerning reality with approximately 60,000 AVs scattered throughout the country [25,26]. This phenomenon could result from the incoherently scattered provisions on AVs management, as shown in Table 1, such as the specific body responsible for the management, the process of handling an AV, and conflict in fines payable for committing obstruction. In the process for AV management, while Act 133 did not specify the length of time before an AV is auctioned or disposed of by other means, Act 333 gives both 12 months and three months under S. 48 and 65, respectively, before cancellation of ownership, seizure, and deregistration. In comparison, the 2020 guidelines require permission from Deputy Public Prosecutor (DPP) and application to the court, which is much more cumbersome, as described in Figure 1.

Similarly, only Act 333 provides voluntary surrender without specifying how the process of surrender should be carried out for proper disposal. Furthermore, none of the legislation specified the disposal procedure for comprehensive management of ELVs, which may arise from either AVs or voluntary surrender. Why a vehicle should be voluntarily surrendered was also not specified. Thus, the provisions were silent on determining an ELV according to age, accident, or inability to maintain. Hence, Malaysia needs comprehensive, clear, and unambiguous legislation for proper ELV maintenance to reduce AV menace. Even though research has proven that legislation alone is insufficient to meet the goals of the letters of the law on AVs [17], awareness raising is vital. In the same vein, it was argued that

raising awareness and developing public interest was germane to the campaign's success against AVs before successful legislative enforcement [18]. With Japan's success [27], raising awareness is an excellent way to achieve success in curbing the AV menace through legislative provisions in Malaysia.

**Table 1.** Review of the existing legislation on AVs in Malaysia.

| Item | Act 133 | Act 333 | 2020 Guidelines |
|---|---|---|---|
| Definition | Obstruction—deposited derelict vehicle Criminal offence S. 46 (1) (e) | Obstruction—leaving a vehicle on the road S. 48 | Idle Vehicles—as defined under S. 46 (1) (e) Act 133 Art. 3.1.1 & 3.2 |
| Process | (a) Remove AVs To a Suitable Place And Detain (b) Owner pays expenses of removal & detention S. 46 (3) (a) & (b) | (a) Clamp the vehicle to a temporary location (b) Send notice of claim within 4 h (c) Vehicle was taken to a permanent place after non-compliance with the notice (d) second notice to claim issued within 24 h requiring claim within one month (e) Vehicle claimed subject to fines (1000-ringgit S. 110) & fees imposed (f) Failure to claim after the expiration of 12 months, DG becomes vested with ownership S. 48 Innovation S. 65 AV removed, a notice of claim is issued requiring claim within three months, failing which a 1month notice shall be Gazetted for auction or other disposals | (a) Local Authority receives, records & tows AV (1 day) (b) Check vehicle status with MySikap portal, PDRM, SSM, and open investigation paper (5 days) (c) AV with criminal problems handed to PDRM, AV without problem deemed completed (d) AV sent to RTD, RTD sent to a department of chemistry for identification (1 day) (e) Issuance of notice of claim to an owner (14 days) (f) CADC meeting (7 days) (g) LOCAL AUTHORITY requests DSP permission to prosecute (1–3 months) (h) LOCAL AUTHORITY request PUSPAKOM to conduct in-situ (1 day) (i) LOCAL AUTHORITY apply for a change of ownership/ cancellation of registration (1 day) (Art. 5) Notice: requisite notice must be sent by LOCAL AUTHORITY to the last known address or left at the usual place of address or business or sent by post. Innovations: The management process shall be concluded within 62–122 days, where there is no legal issue, compared to 6–12 months in Acts 133 & 333. |
| Authorized body | Local Authority (Can delegate power to servants also) | It includes Police, Road transport officer (RTD), Dato Bandar, The Perbadanan Putrajaya, and Highway Authority Malaysia. S. 48 & S. 65 | Local Authority Art. 4.2 |
| Punishment | Fine of 500–1000 ringgit S. 46 (1) (e) | Fine of 1000–5000 ringgit S. 48 | - |
| Disposal | Public auction or other disposal methods S. 116 (1) | (a) Forfeiture S. 48 (b) Public auction or other disposal methods S. 65 | (a) Forfeiture Art. 5.1 (viii) (b) public auction or other disposal methods (c) CAO send to a disposal facility |

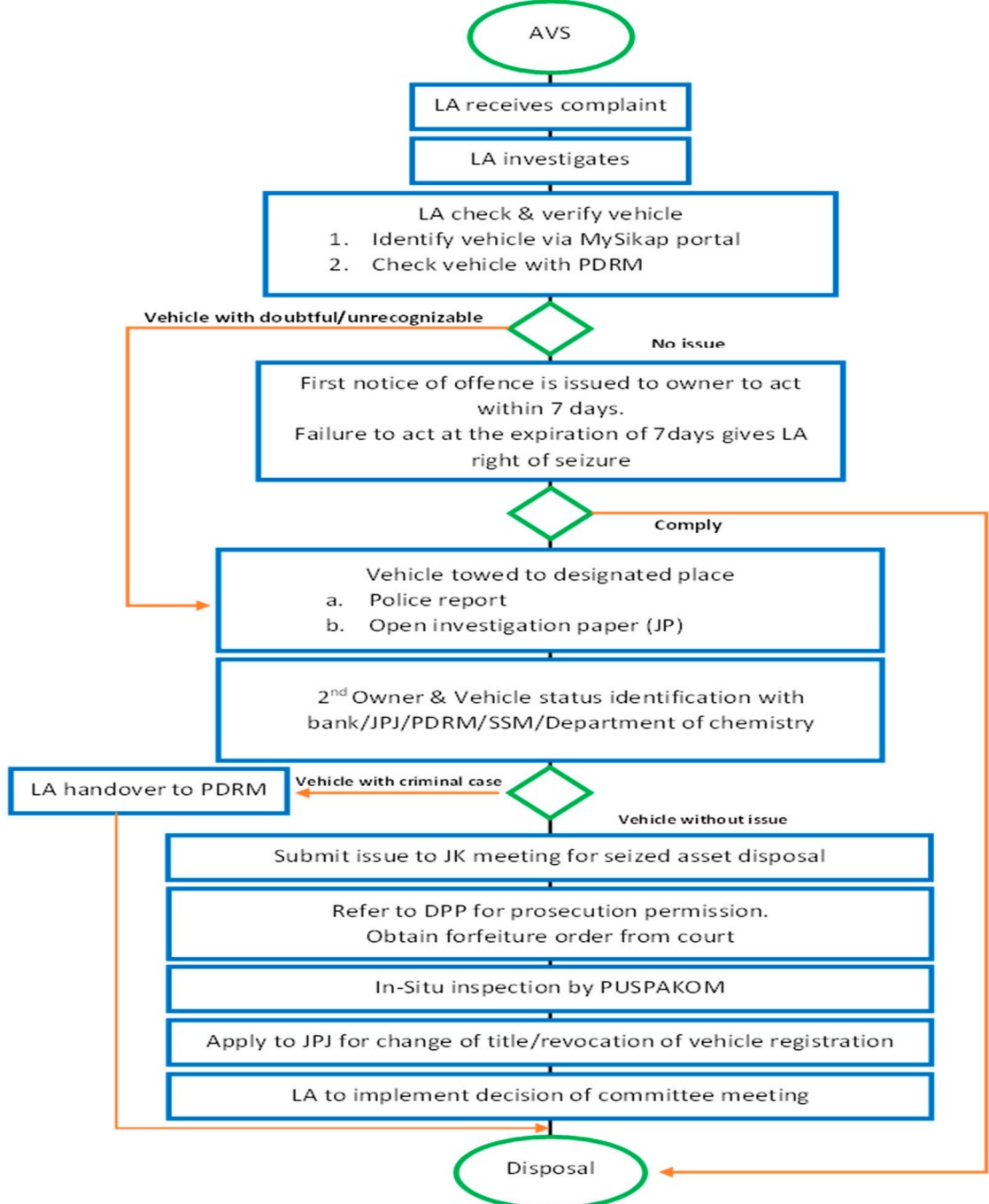

**Figure 1.** 2020 AVs Management Guideline Procedure (Deputy Public Prosecutor, 2020).

The previous significant issues prompted this study, thus we aimed to instigate comprehensive legislation on ELV management for disposal and reuse. The findings should help the government and enforcement officers see the loopholes in the present legislation and why enforcement was not feasible in the circumstance.

## 3. Materials and Methods

In order to analyse the present methods of handling ELVs in Malaysia, the researchers conducted focus group discussions (FGDs) with key stakeholders. These stakeholders included regulators, enforcement agencies, academics, and treatment centre owners, whose industrial insights were deemed relevant to the study. In addition, the participants included

representatives from the Department of Environment, Malaysia Automotive, Robotics and IoT Institute (MARii), local authorities, and the Road Transport Department.

The FGDs were designed to assess the willingness of stakeholders to deal with AVs as the primary ELV issue and determine their level of readiness and direction in regulating ELVs. The discussions were organized and recorded, and agreed-upon questions were posed to guide the conversation. In addition, the discussions were transcribed manually to confirm each speaker's identity during the meetings.

The findings from the FGDs were used to extract critical themes on the AV handling process, which served as a preliminary finding for the study. These themes formed the basis for developing a questionnaire distributed to working citizens and largely middle-class Malaysians who generally own vehicles. The survey was conducted manually and face-to-face with willing respondents.

The questionnaire aimed to gather more information to support the study's conclusion. The responses from the survey were analysed to determine the attitudes and opinions of the respondents regarding the handling of AVs as ELVs. By combining the FGDs and the survey results, the researchers could comprehensively understand the current ELV handling methods in Malaysia. Overall, the FGD and survey questionnaire was made based on previous research by Sitinjak et al. [26], who examined social readiness related to the implementation of ELV management.

### 3.1. Participants

The sample of 40 participants in the study was drawn from a diverse range of automotive stakeholders, including the industry, government, and car owners. The sample size was deemed sufficient to conduct a pilot study, as supported by previous research [28,29]. A detailed demographic analysis revealed that most participants were of Malay ethnicity (38 out of 40), with the remaining two identifying as Chinese.

A gender breakdown was also conducted, showing that 75% of the participants were female, indicating significant ownership of vehicles among females in Malaysia. The educational level of the participants was also considered, with all respondents having at least some form of education, making them suitable for participating in the survey and capable of providing meaningful feedback.

A socioeconomic analysis of the participants was also conducted, revealing that most respondents were government workers (77%) with an average household income of RM 8000 to 11,000. This indicates that they were financially capable of owning and maintaining a vehicle. The demographic and socioeconomic analysis results provide a comprehensive understanding of the population structure and background of the participants.

### 3.2. Data Analysis

The study utilized SPSS Version 27 to analyse the data. The questionnaires used several questions to determine respondents' perceptions of ELV handling in Malaysia. However, the researchers used inductive analysis to allow research findings from the raw data [30]. These raw data were coded, and themes were developed to reduce the possibility of bias [31]. The questionnaire requires respondents' responses from "strongly disagree" to "strongly agree" with 4 points for each item, giving 20 points for analysis. Literature was analysed in the coding process to ensure reliability, and several questions were tested for reliability. The acceptable alpha coefficient is between $0.8 > \alpha \geq 0.7$. Reliability is vital for the total score of 90%, acceptable at 75%, and low at 60%.

## 4. Results

Three key themes emerged with 14 subthemes. These themes were "AVs and their ELVs", "procedure for ELV disposal", and "ELV regulations and dispute resolution". The result is summarized in Tables 2–5.

**Table 2.** AVs and their ELVs–explained.

| No. | Item | Level of Agreement | | | | | Mean | SD | % |
|---|---|---|---|---|---|---|---|---|---|
| | | **1** | **2** | **3** | **4** | **5** | | | |
| 1 | Abandoned vehicles (AVs) and end-of-life vehicles (ELVs) | | | | | | | | |
| | Why do people abandon vehicles? | | | | | | | | |
| High maintenance costs | | 0 | 0 | 14 | 13 | 13 | 3.98 | 0.83166 | 79.6 |
| Badly damaged/accident | | 1 | 0 | 5 | 13 | 21 | 4.33 | 0.88831 | 86.6 |
| Unavailable spare parts | | 2 | 4 | 9 | 17 | 8 | 3.63 | 1.07864 | 72.6 |
| Vehicle no longer in use | | 1 | 1 | 11 | 14 | 13 | 3.93 | 0.97106 | 78.6 |
| Legal validity | | 8 | 3 | 12 | 13 | 4 | 3.05 | 1.28002 | 61 |
| Cumbersome disposal procedure | | 0 | 0 | 10 | 19 | 11 | 4.03 | 0.73336 | 80.6 |
| No knowledge about ELV or disposal rules | | 0 | 0 | 4 | 19 | 17 | 4.33 | 0.65584 | 86.6 |
| Sentiment to vehicle | | 2 | 0 | 4 | 19 | 15 | 4.13 | 0.96576 | 82.6 |
| | How should ELVs be determined? | | | | | | | | |
| According to the age of the vehicle | | 1 | 3 | 7 | 16 | 13 | 3.93 | 1.02250 | 78.6 |
| At the expiration of the warranty | | 1 | 1 | 2 | 22 | 13 | 4.15 | 0.84413 | 83 |
| Yearly periodic inspection | | 9 | 7 | 10 | 8 | 6 | 2.88 | 1.38096 | 57.6 |
| Regular inspection every 2 years | | 6 | 5 | 10 | 12 | 7 | 3.23 | 1.21046 | 64.6 |
| Periodic inspection after 5 years of age | | 1 | 9 | 8 | 17 | 5 | 3.40 | 1.05733 | 68 |
| Periodic inspection after 10 years of age | | 2 | 2 | 4 | 18 | 14 | 4.00 | 1.06217 | 80 |
| At the change in ownership | | 3 | 15 | 4 | 7 | 11 | 3.20 | 1.39963 | 64 |
| | Why should ELVs be determined? | | | | | | | | |
| User safety | | 0 | 0 | 0 | 9 | 31 | 4.78 | 0.42290 | 95.6 |
| User protection against theft | | 0 | 0 | 5 | 15 | 20 | 4.38 | 0.70484 | 87.6 |
| User protection against the structure | | 0 | 0 | 1 | 14 | 25 | 4.60 | 0.54538 | 92 |
| User protection against pollution | | 0 | 1 | 2 | 16 | 20 | 4.38 | 0.74032 | 87.6 |
| User protection against operating system | | 0 | 0 | 1 | 16 | 22 | 4.54 | 0.55470 | 90.8 |
| Preservation of environment | | 1 | 0 | 2 | 18 | 19 | 4.35 | 0.80224 | 87 |
| To determine ELVs | | 1 | 4 | 6 | 16 | 13 | 3.90 | 1.05733 | 78 |
| | What should be done to an ELV? | | | | | | | | |
| Send to licensed disposal centre | | 0 | 0 | 7 | 12 | 21 | 4.35 | 0.76962 | 87 |
| Send to dismantling premises | | 10 | 2 | 6 | 14 | 8 | 3.20 | 1.48842 | 64 |
| Sell as a used car | | 6 | 9 | 7 | 11 | 7 | 3.10 | 1.35495 | 62 |
| No idea | | 6 | 13 | 11 | 5 | 5 | 2.75 | 1.23517 | 55 |
| Leave it at the last location | | 32 | 3 | 2 | 2 | 1 | 1.43 | 0.98417 | 28.5 |

**Table 3.** AVs and their ELVs-description and responsibilities.

| Who Should Remove an AV? | Category | Total Response | % |
|---|---|---|---|
| | Yes | 34 | 85 |
| The owner should send it to the AATF | No | 6 | 15 |
| | Total | 40 | 100 |

**Table 3.** *Cont.*

| Who Should Remove an AV? | Category | Total Response | % |
|---|---|---|---|
| The Local Authority | Yes | 21 | 52.5 |
| | No | 19 | 47.5 |
| | Total | 40 | 100 |
| AATF tows AV | Yes | 6 | 15.0 |
| | No | 34 | 85.0 |
| | Total | 40 | 100 |
| The manufacturer and seller should send it | Yes | 7 | 17.5 |
| | No | 33 | 82.5 |
| | Total | 40 | 100 |
| Insurance company | Yes | 11 | 27.5 |
| | No | 29 | 72.5 |
| | Total | 40 | 100 |
| **When should a vehicle be regarded as AV?** | | | |
| Expiration of road tax | Yes | 34 | 85 |
| | No | 6 | 15 |
| | Total | 40 | 100 |
| Period | 3–4 years | 23 | 65.7 |
| | 5–6 years | 7 | 20 |
| | ≥7 years | 5 | 14.3 |
| | Total | 40 | 100 |
| Desertion in a public area for long | Yes | 36 | 90 |
| | No | 4 | 10 |
| | Total | 40 | 100 |
| Period | 3–4 months | 8 | 21.6 |
| | 5–6 months | 12 | 32.4 |
| | ≥7 months | 17 | 45.9 |
| | Total | 40 | 100 |

**Table 4.** ELVs' disposal processes.

| No. | Item | Level of Agreement | | | | | Mean | SD | % |
|---|---|---|---|---|---|---|---|---|---|
| | | 1 | 2 | 3 | 4 | 5 | | | |
| 2 | ELVs Disposal/Recycle process | | | | | | | | |
| | Surrender | | | | | | | | |
| Voluntary by owner | | 1 | 0 | 5 | 19 | 15 | 4.18 | 0.84391 | 83.6 |
| AATF tows AVs | | 0 | 0 | 4 | 14 | 22 | 4.45 | 0.67748 | 89 |
| Forceful surrender against the owner | | 3 | 6 | 15 | 9 | 7 | 3.28 | 1.15442 | 65.6 |
| Manufacturer | | 3 | 6 | 16 | 9 | 6 | 3.23 | 1.12061 | 64.6 |
| Financial institutions | | 2 | 2 | 17 | 10 | 9 | 3.55 | 1.06096 | 71 |

**Table 4.** *Cont.*

| No. | Item | Level of Agreement | | | | | Mean | SD | % |
|---|---|---|---|---|---|---|---|---|---|
| | | **1** | **2** | **3** | **4** | **5** | | | |
| Deregistration | | | | | | | | | |
| Physical at the road transport department | | 2 | 8 | 11 | 13 | 6 | 3.33 | 1.11832 | 66.6 |
| Online | | 0 | 0 | 3 | 17 | 19 | 4.41 | 0.63743 | 88.2 |
| Disposal costs | | | | | | | | | |
| AATF pay owner | | 0 | 3 | 8 | 17 | 12 | 3.95 | 0.90441 | 79 |
| Owner pays the cost of an ELV | | 3 | 5 | 17 | 0 | 0 | 3.15 | 0.97534 | 63 |
| Buyer should pay at purchase | | 4 | 8 | 16 | 6 | 6 | 3.05 | 1.17561 | 61 |
| Manufacturer should pay | | 3 | 6 | 16 | 9 | 6 | 3.23 | 1.12061 | 64.6 |
| Paid from insurance rebate | | 1 | 0 | 7 | 23 | 9 | 3.98 | 0.80024 | 79.6 |
| Government should bear the cost | | 4 | 4 | 17 | 9 | 6 | 3.23 | 1.20256 | 64.6 |
| Penalty/Incentive | | | | | | | | | |
| Attach incentive to voluntary surrender | | 0 | 0 | 4 | 16 | 20 | 4.45 | 0.67748 | 89 |
| AATF should be given tax exemptions and investment incentives for towing | | 0 | 1 | 9 | 12 | 18 | 4.18 | 0.87376 | 83.6 |
| Advance payment at purchase to be returned after the voluntary surrender | | 2 | 3 | 14 | 11 | 10 | 3.60 | 1.10477 | 72 |
| Fines as a penalty for AV | | 4 | 6 | 10 | 14 | 6 | 3.30 | 1.20256 | 66 |
| Recycling procedure | | | | | | | | | |
| Pickup/surrender | | 0 | 2 | 5 | 15 | 18 | 4.23 | 0.86194 | 84.6 |
| Dismantle vehicle parts | | 0 | 0 | 8 | 16 | 16 | 4.20 | 0.75786 | 84 |
| Destroy vehicle components | | 0 | 0 | 6 | 21 | 13 | 4.18 | 0.67511 | 83.6 |
| Recycle and manufacture reusable parts | | 0 | 0 | 7 | 15 | 18 | 4.28 | 0.75064 | 85.6 |

**Table 5.** ELV regulations and dispute resolution.

| 3 | ELV Regulations and Dispute Resolution | | | | | | | |
|---|---|---|---|---|---|---|---|---|
| Parties Affected by ELV Regulation | | | | | | | | |
| Old/New car owners | 1 | 0 | 3 | 15 | 21 | 4.38 | 0.83781 | 87.6 |
| Owners of AVs | 1 | 2 | 3 | 18 | 16 | 4.15 | 0.94868 | 83 |
| Used car owners | 0 | 1 | 3 | 22 | 14 | 4.22 | 0.69752 | 84.4 |
| Used car owners | 0 | 0 | 3 | 24 | 13 | 4.25 | 0.58835 | 85 |
| Used car sellers | 0 | 2 | 15 | 14 | 9 | 3.75 | 0.86972 | 75 |
| Public transport owners | 0 | 0 | 3 | 24 | 13 | 4.25 | 0.86972 | 85 |
| Car repair shops | 1 | 0 | 12 | 19 | 8 | 3.83 | 0.84391 | 76.6 |
| Shredder Company | 1 | 0 | 9 | 18 | 12 | 4.00 | 0.85896 | 80 |
| Spare parts sellers | 0 | 2 | 10 | 17 | 11 | 3.93 | 0.85896 | 78.6 |
| Financial institutions | 1 | 2 | 22 | 11 | 4 | 3.38 | 0.83781 | 67.6 |

**Table 5.** *Cont.*

| 3 | ELV Regulations and Dispute Resolution | | | | | | | |
|---|---|---|---|---|---|---|---|---|
| | | | Effect of ELV regulations | | | | | |
| High car maintenance costs | 1 | 3 | 5 | 21 | 10 | 3.90 | 0.95542 | 78 |
| Encourage disposal/recycling | 0 | 0 | 6 | 20 | 14 | 4.20 | 0.68687 | 84 |
| Environmentally friendly cars | 0 | 1 | 4 | 23 | 12 | 4.15 | 0.69982 | 83 |
| Solve AVs problems | 0 | 0 | 6 | 20 | 14 | 4.20 | 0.68687 | 84 |
| Guarantee safe-driven cars | 0 | 0 | 7 | 18 | 15 | 4.20 | 0.72324 | 84 |
| ELV part for remanufacturing | 0 | 0 | 12 | 17 | 11 | 3.98 | 0.76753 | 79.6 |
| Dispute Resolution Procedure | | | | | | | Total | % |
| Court Trial | | | | | | | 15 | 37.5 |
| Out-of-court settlement | | | | | | | 25 | 62.5 |

*4.1. Abandoned Vehicles and their End-of-Life*

In Malaysia, AVs are not a new phenomenon. Similar many countries, it is a problem being tackled. This pilot study revealed a decent knowledge about AVs, but no specific and comprehensive legislation exists to curb the menace. Furthermore, from an analysis of the existing legal framework in Table 1, there is no specific period to determine ELV. In order to ascertain the general public's understanding and perception of the appropriate method to deal with and manage AVs, the respondents in the pilot study were asked about the reason for abandoning vehicles, how and why ELV should be determined, when should a vehicle become AV, how to deal with an ELV, and whose responsibility is it to remove an AV to stop obstruction.

According to Table 2, The respondents had a moderate reliability of 75–80% agreement on the subjects of high maintenance cost, badly damaged or accident stricken, unavailability of spare parts, vehicle no longer in use, legal validity, cumbersome disposal procedure, lack of knowledge of ELVs and disposal procedures, and sentiment toward ELV vehicles as reasons for AVs in Malaysia. Recent support findings in Hall and McDonald [17] and Ur [18] listed these factors as reasons for AVs. Because there is no specific provision on when a vehicle can be determined as ELV, the respondents were asked to choose an appropriate option for determining ELVs. Their response revealed reliability support for the expiration of warranty for a vehicle and ten years and more for the inspection of ELV determination with moderate reliabilities of 83% and 80%, respectively. Furthermore, they were firmly against ELV determination at the time of the change of ownership. This inclination means vehicle owners prefer to dispose of their vehicles after ten years rather than a lesser period. Hence, for effective management, the disposal period for ELV determination should begin from at least ten years of age of the vehicle. This is why countries that have succeeded in ELV management chose a period between 10 years and more. For instance, Belgium uses 12–15 years to determine ELV [19], Japan uses 11 years [17,32], and China uses 10 years or 500,000 km [24,33].

According to Table 3, the respondents had a strong reliability acceptance of sending an ELV to an authorized automotive treatment facility (AATF), with 87% choosing to do so rather than abandoning the vehicle at its last location. Furthermore, the respondents showed good knowledge of the benefits of determining an ELV, with an average of 80–92% acceptance rate. These benefits include user safety, protection against theft because of the need to identify the owner before disposal, protection against structure, pollution, operating system, and preservation of the environment. These findings support previous studies that argued that ELVs had economic benefits and environmental and human health risks [7,34]. For proper management and disposal of ELVs, AATFs are the best places to send an ELV [32,35]. When the respondents were asked when a vehicle should be considered abandoned, they seemed to agree strongly with the two periods. The first

was the expiration of road tax, and the second was when a vehicle has been deserted in a public area for a long time. However, their responses seemed to be sceptical and withdrew from specifying the duration of time. This answer revealed a lack of commitment of the respondent in the fight against AVs in Malaysia. Hence, they prefer the owner to send their AV to the AATF with an 85% acceptance rate, compared to the option that either the AV should be towed or a different mode of surrendering the AV should be used.

### 4.2. Procedure for ELV Disposal and Recycling

According to Table 4, the respondents in the study strongly agreed that the procedure for ELV disposal should begin with voluntarily surrendering of the vehicle by the owner or for AATF to tow the AV, with 83.6% and 89% reliabilities, respectively. Consequently, deregistration should be performed with a strong acceptance at 88.2% for online deregistration rather than physically at the Road Transport Department. They also agreed that disposal costs should be paid. However, they seemed comfortable with disposal costs paid to the owner by the AATF or paid from an insurance rebate to the disposal centre with a 79% acceptance rate. Similarly, the respondents had an 89% acceptance rate for attaching incentives to voluntary surrenders and giving the AATF tax exemptions and investment incentives for towing. The reason for this strong reliability of acceptance for incentives would encourage the quick and voluntary surrender of ELVs for disposal and recycling. This result will help address the problem of AVs in Malaysia and help in its management.

Similarly, the respondents had an average 84% reliability acceptance rate for the recycling procedure (Table 4). The process should begin with the surrender and disposal procedure, and then vehicle parts should be dismantled. Non-useful components should be destroyed, and valuable components should be recycled and reused for manufacturing or remanufacturing. This recycling procedure has gained academic support over the years [20,34,35]. Although Japan is successful in ELV management using the "reduce, reuse, and recycle" idea [21], the procedure it adopted is similar to the procedure in this finding [22]. This viewpoint reveals that Malaysia can adopt the procedure for disposal and recycling under legislation to succeed in ELV management. This is similar to Japan's "Home Appliance Recycling L".

### 4.3. Effectiveness of ELV Regulations

This pilot study provides insight into the public perception of ELVs and the inherent weakness in the existing legislation. It was found that the respondents had relatively limited knowledge of ELV regulations and management. They were also not aware of their risks.

This pilot study revealed this anticipated fact [31]. Early research has revealed that allocating resources during the abatement, storage, and deconstruction phases is the most challenging part of ELV disposal [24]. Thus, the first step in ELV management is addressing AV problems [19]. ELV is a potentially political, social, and economic issue requiring careful, calculative, and comprehensive measures to address the menace. Japan, for instance, requests vehicle owners to pay recycling fees for disposal, while manufacturers are responsible for processing ASR [23]. On the other hand, Korea used the Recycling of Electrical and Electronic products and the Automobile Resources Act of 2007 to force manufacturers and importers to collect and recycle ELVs when treatment cost exceeds the ELV value [32].

However, Malaysia has no consistent and comprehensive legislation for the disposal and recycling of ELVs (Table 1). AVs are only treated as traffic obstructions. This research found that Malaysia's most sensitive concerns with automobile owners are being disregarded due to a lack of focus on public readiness to remove AVs. The many reasons why owners choose to abandon their vehicles included breakdown, high cost of towing and repairs, and the inability to afford repairs because owners earn less, and they were evident in this pilot study. These findings support studies in other parts such as Pohnpei and other regions [17,18]. This pilot study also discovered that users might be hesitant to adopt the ELV program if it raises their car ownership costs and shortens their vehicle's lifespan. Following a citizen complaint, the appropriate local authorities will tow the vehicle and

store it at the specified storage facility. The registered owner will be notified immediately as part of the procedure to validate the vehicle's status, as shown in Figure 1.

Securing a forfeiture order from the court is necessary before the Local Authority may proceed with the deregistration. The cars can then be surrendered to the AATF or disposed of differently. However, the process may be prolonged if the disposal requires court approval. As a result, it will remain at the depot until the court proceedings are concluded. Therefore, avoiding the court's order is possible, but only if the registered owner can quickly access courts whenever a conflict arises.

### 4.4. ELV Regulations and Dispute Resolution

Since ELV is not adequately regulated in Malaysia, an attempt to enforce future ELV laws will cause mayhem and receive adverse responses. Similar to some years back, the preliminary step toward ELV policy implementation requiring mandatory annual inspection for vehicles 15 years of age was vehemently opposed and, subsequently, withdrawn [36]. Hence, there is a lack of a comprehensive legislative framework for ELV management [24]. Thus, questions were posed to the respondents directly affected by the ELV regulations. In addition, they were asked about the effect of ELV regulations on the owner's part and the best way to solve conflict or disagreement due to part compliance or non-compliance with ELV regulations.

Table 5 shows that respondents have an average acceptance rate of 70% for persons likely affected by ELV regulations. They include owners of old and new cars, used car owners, owners of AVs, used car sellers, public transport owners, car repair shops, shredder companies, spare parts sellers, and financial institutions. This list presupposes that almost everybody would be affected by ELV regulations. The implication has led Korea in its ELV management process to involve everybody in the country in achieving success [20]. In the same vein, the respondents had an average agreement rate of 80% that ELV regulations affected them in terms of the high car maintenance cost, encouraged disposal and recycling of vehicle components, and, at the same time, caused only environmentally friendly and safe cars are on the roads whilst solving AV problems. They also indicated that remanufacturing is possible with the valuable parts of ELVs. Research across the globe has revealed that ELV management in terms of disposal and recycling has excellent benefits for human and environmental life [7,35]. However, the most exciting part of the survey is that respondents seemed unprepared to the commitment to achieving it the need to address ELVs in Malaysia. This is because most respondents, 62.5% in particular, would go for an out-of-court settlement when a dispute arises following a breach of ELV regulations. Their response shows a readiness to abide by ELV regulations to address ELV problems but a reluctance to accept forceful implementation measures to end ELV problems.

## 5. Discussions and Policy Implications

The following discussion is centred on some critical issues from the findings first. Then, policy implications have been elaborated on.

### 5.1. Efficiency of the Deregistration Process

Table 1 and Figure 1 show that the paperwork required for court approval is cumbersome. As a result, many AVs are abandoned or sold without being deregistered, which may cause future disputes. The research examined the AATF's ability to accept voluntary ELV surrender and address ELV transformational adaptability, as discussed in studies [37,38]. By accepting ELVs directly from the public, the AATF can verify that these vehicles are not AVs, thus resolving the AV issue. In such situations, ELVs could be returned as an alternative to abandonment. For such a surrender to take effect, deregistration procedures must be easily accessible. Hence, the road transport department (RTD) will allow online deregistration only after the owner has personally closed the account.

The pertinent question is, who will transport ELVs to the AATF once deregistration is complete? The owner may drive if the surrender is voluntary, and the car is still working.

In other situations, the car owner can request a towing service, such as the AATF, which will remain on the scene until the car is removed. Any decision may have both legal and practical implications. When it comes to AVs, the situation might become complicated. If there is a financing gap, the subject of who will pay for towing services may arise. The owner, the AATF, or the authorities may contribute to the service in some form. To cash in on each ELV, the owner would most likely want to deposit money into the AATF. However, if the owner is irresponsible and leaves the car unattended, they may not be held accountable. In Malaysia, AATFs agreed to pay for all logistics and transportation costs for the AVs' handling based on a model devised by one of the local authorities, namely, the Kajang Municipal Council (MPKj). This arrangement is a test run with monopoly power for the AATFs.

However, prices cannot be ignored because it will mean that AATFs in the future must be more independent and competitive. The difficulties of comprehensive ELV legislation should not impede AV implementation and voluntary surrenders. The range covers complaints as well as misconduct. More AATFs and inspection centres are needed to provide the services mentioned earlier. Accordingly, speaking with law enforcement, the researchers discovered that municipal governments might hesitate to enforce the relevant regulation strictly. The authorities must juggle the thorny political repercussion through the people's resentment. After removing AVs, they frequently choose not to bill the owner for the abatement costs. Unpaid parking fines may not be collected if the cars are left in public car parks. According to FGD, one of the discussants said that their clients are community members who could not use the van due to financial constraints. Strict enforcement could have unintended repercussions from a political standpoint.

Nevertheless, the pilot study revealed a lack of readiness for strict enforcement. The AATF is now paying for everything out of its pocket, hoping that the scrap metal from the destroyed AVs may one day be enough to pay off the costs. Although AVs present several obstacles to local authorities, the occurrence of legal contradictions is the most serious. Municipal powers can be applied in two ways: via ACT 133 or RTD delegation under ACT 333, as shown in Table 1. However, public opposition to AV bans needs urgent attention.

Vehicle owners, particularly low-income earners, may feel outraged if ELV rules are used to restrict ownership. ELVs may be subject to annual interval checks after a specific age, e.g., ten years, or as stipulated by regulation, to ensure they are still roadworthy. Vehicles that fail the roadworthiness test should be repaired or prohibited from usage. This policy follows because most AVs appear to provide minimal benefit. Scavengers will likely arrive first and grab the beneficial components. Few vehicle owners oppose AV removal since they have sentimental attachments to their cars or other vehicles.

*5.2. Adequacy of the ELV Infrastructure*

In addition to creating an adequate ELV infrastructure, addressing public readiness is vital for dealing with AVs. As a result, Malaysia requires enough AATFs to handle ELVs. Unfortunately, the only AATF service is now limited to the most desired region of Peninsular Malaysia. When the department of environment plan is executed nationally, the number of AATFs could increase to eight in strategically situated places around the country. In addition, a complete ELV system will feature well-regulated inspection centres to ensure that the public uses only safe and environmentally friendly cars. The AV system is riddled with legal inconsistencies despite the recent emergence of legal underpinnings via ministerial guidelines and the earlier deregistration procedure, as shown in Table 1. Many municipal authorities' enforcement departments are unaware of or failure to enforce the legal requirements for disposing of AVs, resulting in the never-ending AVs menace in Malaysia.

*5.3. Roadworthiness and Safety for All*

As awareness of AVs grows, understanding of their severe consequences does also [7,39]. Likewise, if effective legislation is implemented, AVs will disappear. The regulation will ensure that only roadworthy vehicles are used and remove any vehicles unfit from public

roads due to technical faults or inadequate maintenance. Consequently, the number of vehicles that break down in public parking spaces will be minimized. In addition, the ELV legislation will mandate periodic safety and environmental inspections. Since AVs are the major ELV issue, they should be the driving force behind comprehensive ELV regulation. The most significant obstacles stem from the public's unwillingness to accept the possibility that automobile ownership will be financially burdensome. The ELV law could require yearly inspections for older cars to ensure they are safe to drive. Disposal fees should be attached to the time of purchase, as shown in Japan [40], and higher maintenance costs will likely result.

### 5.4. Voluntary Surrender System

Voluntary surrender should address the ELV-related problems outlined in Table 6.

**Table 6.** Voluntary surrender system.

| Stages | With | Without |
| --- | --- | --- |
| Desertion risks | No risks as:<br>- The owner either sends the car to the AATF;<br>- Or worse, the owner may arrange for the car to be collected. | Upon locating a derelict vehicle in a public place, the Local Authority must notify the owner (the owner may not be tracked). |
| Deregistration | The registered owner will deregister the car at the RTD's counter;<br>Or as prescribed (e.g., online). | Securing a forfeiture order from the court is necessary before the Local Authority may proceed with the deregistration (complicated paperwork and court procedure). |
| Abatement | The car will be surrendered to the AATF by the owner or at their request. | Until the owner is duly notified, and a forfeiture order is obtained, the car will be placed at the designated depot (pile-up of vehicles and lack of space). |
| Owners' liabilities, fines, and penalties | The owner absolves from any liability and punishment. Instead, they may receive incentives through payment by the AATF. | - The owner's potential liability includes parking fines, late penalties, and storing charges.<br>- The owner runs the risk of criminal charges. |
| Legal risks | Almost nil. | The owner retains the legal right to bring any action against the mishandling of their property. The omission of deregistration can cause the illegal reuse of the registered number by criminal or irresponsible parties. |

The Voluntary Surrender System for ELVs is designed to manage ELVs in a sustainable and environmentally friendly manner. It enables vehicle owners to safely dispose of their old cars, reducing the potential environmental harm caused by improper disposal. Moreover, the system also helps to conserve valuable resources, such as metals and plastics, that can be recycled or reused in other industries [41].

This system plays a significant role in addressing the automotive industry's waste management challenges. Ensuring the responsible disposal of ELVs helps reduce vehicle waste's environmental impact and promotes sustainability. Additionally, the system provides a valuable opportunity to recover valuable resources, which can be used to create new products and support the growth of various industries.

In conclusion, the Voluntary Surrender System in ELV is an essential development to create a more sustainable and environmentally responsible automotive industry. Its implementation promotes the safe and responsible disposal of ELVs, reduces waste, and conserves valuable resources. The system is a positive step towards creating a greener and more sustainable future for the automotive industry and the environment.

### 6. Conclusions

Leaving abandoned metal items in the environment will destroy the environment and be hazardous to the human body because it will enter the body through the food chain

and cause health hazards. The study sought to add empirical rigour to discussions on the issue of AVs by approaching it from ELV regulatory perspective and highlighting that an effective voluntary surrender system is a stepping stone toward holistic ELV regulation. Voluntary surrender may be customized to suit a more extensive agenda of environment conservation; as such, this is what ELV regulation should aim for. First, however, the regulation should prescribe the AATF as the endpoint for the surrendered vehicles. This expectation is because the AATF provides an environmental disposal process not offered by other disposal centres.

The importance of recycling and utilizing used cars cannot be overemphasized. First, a considerable resource utilization potential stems from recycled or remanufactured components. Secondly, it has the potential to save energy and reduce emissions of greenhouse gases. It also protects the environment from the contamination of the mining and refining of the original mineral reserves. Thirdly, it potentially lowers unemployment rates because implementing recycling programs for used automobiles can result in new enterprises needing an increased workforce. Finally, it reduces the cost of the products. The extracted items can be bought for low prices, and the quality of the goods can be improved without spending too much money.

Malaysia needs to effectively address the issue of AVs and their harmful environmental effects. Therefore, it is timely to consider using a surveillance detector to identify AV owners who have refused to relinquish their vehicles voluntarily. The detector can ensure the reinfusing department access, identification, and effective management through the model of an abandoned object detector or binocular information reconstruction and recognition, a 3D surveillance detector used in road traffic surveillance.

This study had several flaws, including that the researchers did not scale the acceptance of existing ELV legislation into the study and that it was a preliminary study with few respondents who could not reflect the entire population. However, this research can also serve as a foundation for future research by including additional respondents to obtain results that can be generalized, as well as the addition of measuring indicators to obtain more thorough results.

**Author Contributions:** Conceptualization, H.M.A. and J.Z.H.; methodology, R.M.K., C.S. and M.H.M.S.; software, R.I.; validation, H.M.A., R.I. and J.Z.H.; formal analysis, G.E.K.; investigation, G.E.K.; resources, C.S.; data curation, G.E.K.; writing—original draft preparation, G.E.K., M.H.M.S., C.S. and R.M.K.; writing—review and editing, V.S.; visualization, V.S.; supervision, V.S., H.M.A. and R.I; project administration, M.H.M.S.; funding acquisition, H.M.A. All authors have read and agreed to the published version of the manuscript.

**Funding:** Ministry of Higher Education (Malaysia) TRGS/1/2020/UKM/02/1/3.

**Institutional Review Board Statement:** The research project with ethical approval number UKM PPI/111/8/JEP-2021-595 has been reviewed and approved by the Institutional Review Board at Universiti Kebangsaan Malaysia. The IRB ensures that the study adheres to ethical standards for research involving human subjects, as outlined in the Declaration of Helsinki and local regulations.

**Informed Consent Statement:** Not applicable.

**Data Availability Statement:** The data that support the findings of this study are available upon request from the corresponding authors.

**Acknowledgments:** The authors would like to thank and appreciate the Ministry of Higher Education (Malaysia) for providing the research funding under TRGS/1/2020/UKM/02/1/3.

**Conflicts of Interest:** The authors declare no conflict of interest.

## Appendix A. Summary of Previous Studies on ELV

| Author(s) and Year | Methods | AV and ELV | | DP and RL | | RG and DR | | ER | ST | Country |
|---|---|---|---|---|---|---|---|---|---|---|
| | | AV | ELV | DP | RL | RG | DR | | | |
| (Yu et al., 2022) [1] | Review | √ | √ | √ | √ | | | √ | √ | China |
| (Goggin, 2019) [2] | Theoretical | √ | | | | | | | √ | Australia |
| (Wang et al., 2015) [3] | Experiment | √ | | √ | √ | | | √ | | China |
| (Muni et al., 2021) [4] | Digital data | √ | √ | | | | | √ | | USA |
| (Ji et al., 2017) [5] | Survey | √ | √ | | | | | √ | | China |
| (Da Costa et al., 2022) [6] | Questionnaire | | √ | √ | √ | | | √ | √ | Brazil |
| (Modoi and Mihai, 2022) [7] | Case Study | | √ | √ | √ | √ | √ | √ | √ | Romania |
| (Kurogi et al., 2021) [8] | Quantitative | √ | √ | √ | √ | | | √ | √ | Japan |
| (Shahriarirad et al., 2020) [9] | Survey | | | | √ | | | √ | √ | Iran |
| (Chaabane et al., 2021) [10] | Analysis | | √ | √ | √ | | | | √ | Canada |
| (Restrepo et al., 2020) [13] | MFA Analysis | √ | √ | √ | | √ | | √ | √ | Switzerland |
| (Jawi et al., 2016) [37] | Survey | | √ | | | √ | | √ | | Malaysia |
| (Raja Mamat et al., 2016) [15] | Survey | √ | | | | | | | √ | Malaysia |
| (Hall and McDonald, 2020) [17] | Theoretical | √ | √ | √ | | √ | | | √ | Japan |
| (Hafiz Ur, 2015) [18] | Theoretical | √ | √ | | | √ | | | | Japan |
| (Nawawi et al., 2020) [16] | Questionnaire | | √ | √ | | | | √ | | Malaysia |
| (Inghels et al., 2016) [19] | System Dynamics | √ | √ | √ | √ | √ | | √ | √ | Belgium |
| (Jang et al., 2022) [20] | MFA Analysis | | √ | √ | √ | √ | | | √ | Korea |
| (Zhou et al., 2019) [21] | Review | | √ | √ | √ | √ | √ | √ | √ | China |
| (Rastogi et al., 2020) [22] | Analysis | | √ | √ | √ | √ | | √ | | India |
| (Prakash and Kumar, 2020) [23] | Experiment | | √ | √ | √ | | | √ | | India |
| (Harun et al., 2021) [24] | Survey | | √ | √ | √ | √ | | | | Malaysia |
| (Ismail et al., 2023) [25] | Survey | √ | √ | | | √ | | | | Malaysia |
| (Sitinjak, I., T. et al., 2022) [26] | Survey | √ | √ | | | √ | | √ | | Malaysia |
| (Susmono, 2017) [27] | Review | | √ | √ | √ | √ | | | | Indonesia |
| (Monteblanco and Vanos, 2021) [31] | Pilot | | √ | | | | | √ | | USA |
| (Zuccaro et al., 2019) [32] | Experiment | | √ | √ | | | | | √ | UAE |
| (Sitinjak, I., B. et al., 2022) [33] | Survey | | √ | | | √ | | | | Malaysia |
| (Gan and Luo, 2017) [34] | Analysis | | √ | | √ | | | √ | | China |
| (Henk et al., 2021) [35] | Review | | √ | √ | √ | √ | √ | √ | √ | Finland |
| (Jin et al., 2022) [38] | Comparative Analysis | | √ | √ | √ | | | √ | √ | Japan |
| (Singh et al., 2019) [39] | Review | | √ | √ | √ | √ | | | | India |
| (Zhao and Chen, 2011) [11] | Comparative Analysis | | √ | √ | √ | √ | | | √ | China |
| (Ning, 2020) [12] | Analysis | | √ | √ | √ | √ | | √ | √ | China |
| (Sitinjak, I., B., F., and S., 2022) [36] | Survey | | √ | | | √ | | √ | | Malaysia |
| (Simic, 2019) [41] | Case Study | | √ | √ | | √ | | | | Serbia |

AV = Abandoned Vehicles; ELV = End-of-Life of Vehicles; DP and RL = Disposal and Recycling; RG and DR = Regulation and Dispute Resolution; DP = Disposal; RL = Recycle; RG = Regulations; DR = Dispute Resolution; ER = Environmental Risk; ST = Sustainability.

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
