# Peer review of "Exploring Public Perceptions and Disposal Procedures in the Development of a Comprehensive End-of-Life Vehicle Regulation in Malaysia: A Pilot Study"

_sustainability, doi:10.3390/su15064786_

Round 1

Reviewer 1 Report

This paper studied the public view on the disposal of end-of-life vehicles in Malaysia. This study is interesting. My comments to improve the paper further are as follows: 

The title of the paper needs to match its content better. This paper is mainly focused on the public views; not on how to improve regulations.

At the beginning of section 2, it is mentioned that a focus group discussion with key stakeholders was held. However, not too much detail about the discussion is presented in the paper. If possible, the authors could add more detail such as opinions, conclusions, suggestions, and lessons learned from the discussion.

The public view survey alone may not be sufficient to support some of the discussions and recommendations in section 4. Take the proposed voluntary surrender system for example, it not only needs input from the general public but also needs input from industry and government.

The paper has some typos (e.g., SPPS version 27 in line 19) that need to be addressed.

Author Response

*please check the blue text in the text

Reviewers' Comments to the Authors: 

Reviewer 1

  1. The title of the paper needs to match its content better. This paper is mainly focused on the public views, not on how to improve regulations.

Author response: Thank you for pointing this out. The reviewer is correct, and we have changed the title as follows: “Exploring Public Perceptions and Disposal Procedures in the Development of a Comprehensive End-of-Life Vehicle Regulation in Malaysia: A Pilot Study”.

  1. At the beginning of section 2, it is mentioned that a focus group discussion with key stakeholders was held. However, not too much detail about the discussion is presented in the paper. If possible, the authors could add more detail, such as opinions, conclusions, suggestions, and lessons learned from the discussion.

Author response: As suggested by the reviewer, we have re-engineered our method following the reviewers' suggestions. Please see section 2:

The researchers conducted focus group discussions (FGDs) with key stakeholders to analyze the present methods of handling ELVs in Malaysia. These stakeholders included regulators, enforcement agencies, academics, and treatment center owners, whose industrial insights were deemed relevant to the study. In addition, the participants included representatives from the Department of Environment, Malaysia Automotive, Robotics & IoT Institute (MARii), local authorities, and the Road Transport Department.

The FGDs were designed to assess the willingness of stakeholders to deal with AVs as the primary ELV issue and determine their level of readiness and direction in regulating ELVs. The discussions were organized and recorded, and agreed-upon questions were posed to guide the conversation. In addition, the discussions were transcribed manually to confirm each speaker's identity during the meetings.

The findings from the FGDs were used to extract critical themes on the AV handling process, which served as a preliminary finding for the study. These themes formed the basis for developing a questionnaire distributed to working citizens and largely middle-class Malaysians who generally own vehicles. The survey was conducted manually and face-to-face with willing respondents.

The questionnaire aimed to gather more information to support the study's conclusion. The responses from the survey were analyzed to determine the attitudes and opinions of the respondents regarding the handling of AVs as ELVs. By combining the results from the FGDs and the survey, the researchers could comprehensively understand the current state of ELV handling methods in Malaysia.

  1. The public view survey alone may not be sufficient to support some of the discussions and recommendations in section 4. Take the proposed voluntary surrender system for example, it not only needs input from the general public but also needs input from industry and government.

Author response: Thank you for taking attention to this point; we have conveyed that this is our pilot study data, where the results will form the basis for our subsequent study. For our respondents, we have taken all automotive stakeholders; this includes stakeholders and the automotive industry in Malaysia.

  1. The paper has some typos (e.g., SPPS version 27 in line 19) that need to be addressed

Author response: Thank you for pointing this out. We have rechecked and changed several typos in our work.

Reviewer 2 Report

Overall, the manuscript addresses a relevant topic, although supplementations and improvements are required as well as changes in the article structure. That is why, I would like to list of major and minor remarks below. I suggest ‘rebuilding’ the article by giving a proper structure (without mixing results, discussion, introduction and literature analysis) and improving the quality of results presentation. In this form it is difficult to be understood by the audience. After that it should be resubmitted to the journal.

General remarks (major):

- The ‘Introduction’ part is too long and too deep with some parts written as research results (after analysing case studies – e.g. lines 73 – 88). The author(s) should divide it into parts and concentrate on research background, main problems, hypothesis and objectives;

- Literature review is not a separate section. Due to this at least a table summarizing previous studies concerning the topic should be supplemented. This would highlight the novelty of this study;

- Due to the fact that the article present results of the preliminary study the author(s) should justify selection of the group of respondents and the content of the questions asked. Is this study inspired by other research or recommended after previous studies?

- Part ‘2.1. Participants’ is not ‘deep’ enough. The author(s) do not use schemes which are more than appropriate to show the population structure. It should be improved;

- Research results are placed in the ‘Discussion’ part (Table 5), that is why there is an impression that the results are presented randomly;

- There is only 1 sentence in ‘4.5. Voluntary Surrender System’ part and Table 6 needs to be referenced more.

Details (minor):

- line 45 – are these whole titles of legal acts? – if not, they should be addressed properly;

- line 70 – supplement the date (at least a year) of ‘Convention’;

- Table 1 – is it an own elaboration and part of the desk research or taken from another source? should be highlighted;

- Skip words such as ‘above’ and ‘below’ – finally editors decide where tables and figures should be placed and giving the reference by pointing the number is enough;

Author Response

Please check a green text on document

Reviewer 2

  1. The 'Introduction' part is too long and too deep with some parts written as research results (after analysing case studies – e.g. lines 73 – 88). The author(s) should divide it into parts and concentrate on research background, main problems, hypothesis and objectives;

Author response: Thank you for the comment. This part has been reduced by 41% to concentrate on the research background, main problems, hypothesis, and objectives, as suggested by the second reviewer.

  1. Literature review is not a separate section. Due to this at least a table summarizing previous studies concerning the topic should be supplemented. This would highlight the novelty of this study;

Author response: To better highlight the novelty of this study we have added “Review of ELV Regulation and Management in Malaysia”. A table summarizing existing legislation has been provided to demonstrate the need for the development of comprehensive ELV regulation in Malaysia.

  1. Due to the fact that the article present results of the preliminary study the author(s) should justify selection of the group of respondents and the content of the questions asked. Is this study inspired by other research or recommended after previous studies?

Author response: Thank you for your comment, this issue has been clarified as follows: “Overall the questionnaire made in the FGD and survey was made based on previous research by Sitinjak et al. [26] who examined social readiness related to the implementation of ELV management.”.

  1. Part' 2.1. Participants' is not 'deep' enough. The author(s) do not use schemes which are more than appropriate to show the population structure. It should be improved;

Author response: Thank you for pointing this out. The reviewer is correct, and we have expanded part 2.1 deeper base on the reviewer's comment.

The sample of 40 participants in the study was drawn from a diverse range of automotive stakeholders, including the industry, government, and car owners. The sample size was deemed sufficient to conduct a pilot study, as supported by previous research [28], [29]. A detailed demographic analysis of the participants was conducted, revealing that most participants were of Malay ethnicity (38 out of 40), with the remaining two identifying as Chinese.

A gender breakdown was also conducted, showing that 75% of the participants were female, indicating significant ownership of vehicles among females in Malaysia. The educational level of the participants was also considered, with all respondents having at least some form of education, making them suitable for participating in the survey and capable of providing meaningful feedback.

A socioeconomic analysis of the participants was also conducted, revealing that the majority of respondents were government workers (77%) with an average household income of 8,000 to 11,000 ringgit. This indicates that they were financially capable of owning and maintaining a vehicle. The demographic and socioeconomic analysis results provide a comprehensive understanding of the population structure and background of the participants.

  1. Research results are placed in the 'Discussion' part (Table 5), that is why there is an impression that the results are presented randomly;

Author response: Thank you for your valuable comment. We have moved the research result (see point 3.3).

  1. There is only 1 sentence in '4.5. Voluntary Surrender System' part and Table 6 needs to be referenced more.

Author response: Thank you for pointing this out. We agree with this comment. Therefore, we have added a discussion on that point as follows: “The Voluntary Surrender System in ELV is designed to manage end-of-life vehicles sustainable and environmentally friendly manner. It enables vehicle owners to safely dispose of their old cars, reducing the potential environmental harm caused by improper disposal. Moreover, the system also helps to conserve valuable resources, such as metals and plastics, that can be recycled or reused in other industries.

This system plays a significant role in addressing the waste management challenges faced by the automotive industry. Ensuring the responsible disposal of ELVs helps reduce vehicle waste's environmental impact and promotes sustainability. Additionally, the system provides a valuable opportunity to recover valuable resources, which can be used to create new products and support the growth of various industries.

In conclusion, the Voluntary Surrender System in ELV is an essential development to create a more sustainable and environmentally responsible automotive industry. Its implementation promotes the safe and responsible disposal of end-of-life vehicles, reduces waste, and conserves valuable resources. The system is a positive step towards creating a greener and more sustainable future for the automotive industry and the environment.”.

  1. line 45 – are these whole titles of legal acts? – if not, they should be addressed properly;

  1. line 70 – supplement the date (at least a year) of 'Convention';

Author response: Thank you for the comment. This has been supplemented in the text on line 85.

  1. Table 1 – is it an own elaboration and part of the desk research or taken from another source? should be highlighted;

Author response: This table summarized the existing legislation on AVs In Malaysia. It is a part of the desk research to demonstrate the need for the development of comprehensive ELV regulation in Malaysia.

  1. Skip words such as 'above' and 'below' – finally editors decide where tables and figures should be placed and giving the reference by pointing the number is enough;

Author response: Thank you for pointing this out. We agree with this comment. We have deleted all words 'above' and 'below'

Reviewer 3 Report

The study focuses on the Instigating Regulations on Abandoned Vehicles: A Pilot Study 2 in Malaysia.

1. In the abstract, is it SPPS or SPSS? The acronyms should be defined in the abstract.

2. The key findings of the study is lacking in the abstract.

3. In line 29 "..vehicles and other electronic products..." Are vehicles electronic products?

4. In line 40-44, since the study focus on Malaysia, it will be appropriate to use data from Malaysia.

5. Lines 68-69 should be referenced.

6. How abandoned vehicles posed environmental risk should be highlighted in the introduction.

7. Lines 104-105 need to be revised for clarity.

8. In line 122, what is DPP?

9. Figure 1 needs to be properly explain. What is the source?

10. The discussion and recommendation should be tied to the motivations/objectives of the study.

11. The conclusion  should not be referenced. The authors should propose a policy implication of the study.

Author Response

Please check the brown text on the attachment

Reviewer 3

  1. In the abstract, is it SPPS or SPSS? The acronyms should be defined in the abstract.

Author response: Thank you for pointing this out. We have changed SPPS to SPSS.

  1. The key findings of the study is lacking in the abstract.

Author response: Thank you for your suggestion; we have made the following changes: “The growing global demand for new cars has resulted in a rise in end-of-life (ELV) vehicles, both with usable and non-usable parts. Malaysia faces a particularly pressing issue of abandoned vehicles (AVs), as the country currently lacks comprehensive legislation for managing ELV. In comparison, countries such as the US, Japan, Belgium, Korea, and China have well-established ELV management policies. In light of this, a pilot study was conducted to assess the public's perception and attitudes toward ELV issues and regulations in Malaysia. The study aimed to gather information about participants' views on surrender, deregistration, disposal costs, incentives, recycling, and ELV regulations. The questionnaire used in the study was based on the findings from a previous report. The data was analyzed using SPSS version 27, based on 40 interviews conducted with a diverse range of participants. The results of the study revealed three primary themes: "AV and its ELV," "ELV disposal procedures," and "ELV regulation and conflict resolution," with 14 sub-themes. The study adopts a regulatory approach towards ELV and AVs and highlights the importance of an effective voluntary surrender system as a step towards comprehensive ELV regulation in Malaysia. The findings of this pilot study provide valuable insights into the Malaysian public's views on ELV and can inform the development of more effective and comprehensive ELV legislation.”.

  1. In line 29 "..vehicles and other electronic products..." Are vehicles electronic products?

Author response: Thank you for pointing this out. We are focused on vehicles, so we decided to omit other electronic products.

  1. In line 40-44, since the study focus on Malaysia, it will be appropriate to use data from Malaysia.

Author response: We think this is an excellent suggestion. We use other data because in Malaysia, there are still few studies on ELV, and there are no practices related to ELV. We take examples from other countries.

  1. Lines 68-69 should be referenced.

Author response: Thank you for pointing this out. We have added a reference for that statement.

  1. How abandoned vehicles posed environmental risk should be highlighted in the introduction.

Author response: Thank you for pointing this out. We agree with this comment. We have added some explanation regarding how abandoned vehicles pose an environmental risk: “The presence of abandoned vehicles (AVs) in the environment poses significant ecological risks, including soil and groundwater contamination, air pollution, depletion of valuable land resources, and aesthetic degradation of the visual environment. The leakage of toxic substances, such as lead-acid batteries, fuels, oils, coolants, and chemicals, from AVs can result in soil and water pollution, posing a threat to the local ecosystem, wildlife, and human health. Additionally, the accumulation of AVs in landfill sites and the release of hazardous air pollutants and greenhouse gases can exacerbate environmental degradation. Thus, it is imperative to develop and implement effective management and disposal practices to mitigate the ecological impacts of AVs [13].”.

  1. Lines 104-105 need to be revised for clarity.

Author response: Thank you for comment. This part has been revised for clarity, i.e., “Although Table 1 indicates the existence of clear provisions for the management of abandoned vehicles (AVs), recent data reveals a concerning reality with approximately 60,000 AVs scattered throughout the country.”.

  1. In line 122, what is DPP?

Author response: Thank you for this point. This abbreviation has been added in the text as follows: “DPP is Deputy Public Prosecutor (DPP)”.

  1. Figure 1 needs to be properly explain. What is the source?

Author response: Thank you for pointing this out. We have added resources for Figure 1.

  1. The discussion and recommendation should be tied to the motivations/objectives of the study.

Author response: Thank you for the feedback. Section 5 named “Discussions and Policy Implications” has been rewritten and extended to better follow the motivations/objectives of the study.

  1. The conclusion should not be referenced. The authors should propose a policy implication of the study.

Author response: Thank you for pointing this out. We have deleted all references from the conclusion. Policy implications have been elaborated on in Section 5 named “Discussions and Policy Implications”.

Round 2

Reviewer 1 Report

The authors have improved the paper and adequately addressed my previous comments. I have no more comments. 

Author Response

I wanted to reach out to thank you for your insightful feedback on our manuscript, Exploring Public Perceptions and Disposal Procedures in the Development of a Comprehensive End-of-Life Vehicle Regulation in Malaysia: A Pilot Study. Your thorough review and positive feedback have been invaluable in helping us refine and strengthen our work.

We greatly appreciate your support for our research and are thrilled that you found our study to be valuable and significant. We are confident that our findings will contribute to the field.

We are grateful for your advocacy of our work and hope that you will recommend its publication. Thank you again for your time and thoughtful comments

Reviewer 2 Report

Thank you for your careful revisions. Most of my comments from first round review were addressed. The paper has a significant improvement and the quality of the structure much higher.

However, I have 1 minor remark to your improved version of the article (the suggested supplementation has not been made so far):

A table or scheme summarizing previous studies concerning the topic should be supplemented. This would highlight the novelty of this study.

Author Response

Dear,

I would like to extend my sincere gratitude for your thoughtful review of our manuscript, Exploring Public Perceptions and Disposal Procedures in the Development of a Comprehensive End-of-Life Vehicle Regulation in Malaysia: A Pilot Study. Your feedback has been invaluable in helping us refine and improve our study.

We are thrilled to hear that you found our research to be valuable and significant. In response to your comments and suggestions, we have made several updates and revisions to the manuscript, including specific changes made to the manuscript (see blue one and appendix 1)

We hope that these changes address any concerns you had and demonstrate our commitment to rigor and clarity in our research. We appreciate your support for our work and are grateful for your advocacy for its publication.

Thank you again for your time and thoughtful comments.

Best regards,

Charli Sitinjak

Reviewer 3 Report

The authors have addressed the comments raised. I therefore recommend the manuscript for acceptance. 

Author Response

Dear,

I would like to express my sincere gratitude for your thorough review of our manuscript, Exploring Public Perceptions and Disposal Procedures in the Development of a Comprehensive End-of-Life Vehicle Regulation in Malaysia: A Pilot Study. Your insightful comments and constructive feedback have been immensely helpful in improving the quality and rigor of our research.

We appreciate your support for our work and are pleased that you found our study to be interesting and worthwhile. We understand that you had some concerns regarding ELV issue and we have carefully considered and addressed these concerns in our revised manuscript.

Thank you again for your time and valuable feedback.